# The effect of common paralytic agents used for fluorescence imaging on redox tone and ATP levels in *Caenorhabditis elegans*

**Katherine S. Morton**[ORCID]ᵒ, **Ashlyn K. Wahl**ᵒ, **Joel N. Meyer***

Nicholas School of Environment, Duke University, Durham, North Carolina, United States of America

ᵒ These authors contributed equally to this work.
* Joel.meyer@duke.edu

**Data Availability Statement:** All relevant data are within the manuscript and its Supporting Information files.

## Abstract

One aspect of *Caenorhabditis elegans* that makes it a highly valuable model organism is the ease of use of *in vivo* genetic reporters, facilitated by its transparent cuticle and highly tractable genetics. Despite the rapid advancement of these technologies, worms must be paralyzed for most imaging applications, and few investigations have characterized the impacts of common chemical anesthetic methods on the parameters measured, in particular biochemical measurements such as cellular energetics and redox tone. Using two dynamic reporters, QUEEN-2m for relative ATP levels and reduction-oxidation sensitive GFP (roGFP) for redox tone, we assess the impact of commonly used chemical paralytics. We report that no chemical anesthetic is entirely effective at doses required for full paralysis without altering redox tone or ATP levels, and that anesthetic use alters the detected outcome of rotenone exposure on relative ATP levels and redox tone. We also assess the use of cold shock, commonly used in combination with physical restraint methods, and find that cold shock does not alter either ATP levels or redox tone. In addition to informing which paralytics are most appropriate for research in these topics, we highlight the need for tailoring the use of anesthetics to different endpoints and experimental questions. Further, we reinforce the need for developing less disruptive paralytic methods for optimal imaging of dynamic *in vivo* reporters.

## Introduction

*Caenorhabditis elegans* is a powerful model organism widely used by researchers to study developmental biology, neurobiology, metabolism, and human diseases such as cancer, metabolic disorders, and neurodegeneration [1,2]. Worms reach adulthood in just 3 days, are highly genetically tractable, and are inexpensive to maintain. Of particular utility for *C. elegans* researchers is their transparency, making them ideal for microscopy-based investigations. Ranging from assessing localization and morphology, to gene expression via production of gene-specific promoter-driven fluorescent "reporter" proteins, to dynamic real time read-outs of biochemical endpoints such as $Ca^{2+}$ dynamics, redox tone, and ATP levels, light microscopy

**Funding:** This study was funded by the National Intitutes for Health (https://www.nih.gov/; JM-P42ES010356, R01ES034270,2R44OD024963, P40 OD010440, and T32ES021432). The funders had no role in study design, data collection and analysis, decision to publish, or preparation of the manuscript.

**Competing interests:** The authors have declared that no competing interests exist.

is at the forefront of methods used in *C. elegans* research [3]. The generation of real-time *in vivo* reporters that allow for fluorescent readouts of biochemical endpoints offers large advantages over traditional biochemical assessments, including cell-specific measurements, but their utility may be limited if the chemical paralytics typically used for imaging alter the biochemical endpoints being measured. Unlike transcriptional reporter strains, these sensors respond rapidly to intracellular conditions, including changes that may be introduced by anesthetics. Two areas of sensor development that have driven significant advancement aim to assess reactive oxygen species or redox tone and ATP levels.

Reactive oxygen species (ROS) are various products of the reduction of oxygen from the addition of electrons; they serve necessary signaling roles at basal levels, but at high levels they can oxidize DNA, proteins, and lipids resulting in cellular damage [4]. In most cells, mitochondria are the primary producers of ROS via the electron transport chain [4]. Historically, detecting alterations to reactive oxygen species production in *C. elegans* has been achieved through quantification of GFP driven by promoters of antioxidant genes, quantifying byproducts of oxidation (oxidized lipids, DNA damage), or the use of dyes [5]. However, these methods can lack specificity and sensitivity, are difficult to quantify in individual compartments, or are not dynamic enough to demonstrate rapid changes in redox tone. More recently, fluorescent sensors such as Hyper and reduction:oxidation sensitive GFP (roGFP) have allowed for compartment-specific expression and assessment of response in real time [6].

Similarly, cellular regulation of ATP levels can now be dynamically monitored *in vivo* through reporters such as PercevalHR and QUEEN-2m [7,8]. Produced through glycolysis and oxidative phosphorylation, ATP levels and ATP:ADP ratio supply valuable insight into bioenergetic function and can indicate differences in metabolite availability, TCA cycle and glycolytic function, and overall cellular bioenergetic homeostasis. In the absence of *in vivo* reporters, ATP must be measured quantitatively at the whole worm level, or after cell sorting. Dynamic by design, ATP levels change rapidly rendering analyses that require more invasive methods less accurate. The use of fluorescent reporters bypasses this issue at the cost of requiring paralysis or immobilization for effective imaging.

Though critical to accurate reporting and interpretation of these dynamic fluorescent sensors, no document currently exists that comprehensively reports the effects of common anesthetics on *C. elegans* cellular redox tone and ATP levels. Whether mounted on slides or imaged in multiwell plates, most common imaging techniques require the full paralysis of nematodes to be effective. Historically, and most commonly today, this is conducted with chemical paralytic agents. As each anesthetic has a discrete mechanism by which paralysis is induced, we evaluated 5 different chemical paralytic agents as well as cold shock.

Likely the most common paralytic, sodium azide is a well-established inhibitor of mitochondrial respiration that blocks cytochrome c oxidase (electron transport chain Complex IV) by binding to the oxygen reduction site [9,10]. Paralysis from exposure to sodium azide is generally ascribed to the rapid depletion of ATP necessary for movement. Azide also inhibits ATP hydrolysis by $F_1$-ATPases, leading to stimulation of potassium channels in bovine *in vitro* models [9]. ATP-gated potassium channels are essential for coupling cellular ATP levels to membrane excitability and represent a potential contribution to both paralysis and off-target effects [11]. Sodium azide use in *C. elegans* was previously not found to induce the stress response pathways associated with *hif-1*, *TMEM-135*, *hsp-4*, *hsp-16.2*, or *gcs-1* [12]. It has been used as a paralytic agent for a wide range of studies, including neurotoxicology, gravitaxis and oxidative stress [13–16].

Levamisole HCl has long been used as an anti-helminthic and is used in *C. elegans* not only as a paralytic, but to examine anthelminthic resistance. Levamisole binds to and activates L-

type acetylcholine receptors leading to sustained $Ca^{2+}$ flux into neurons and muscles, ultimately inducing spastic paralysis [17–19].

2,3-Butanedione monoxime (2,3-BDM) is an inhibitor of skeletal muscle myosin II, that specifically inhibits the myosin ATPase activity. This inhibition causes a decrease in muscle force production [20]. A recent investigation of the contribution of the reproductive system to oxygen consumption in *C. elegans* utilized 2,3-BDM, stating that this method of inhibition is "not increasing ATP synthesis or mitochondrial oxygen consumption" [21]; however, to our knowledge, no studies have empirically tested the effect of 2,3-BDM on ATP levels. 2,3-BDM induces flaccid paralysis, such that muscles are not contracted during imaging, but this effect appears to be relatively slow; previous work often allows the worms a full hour to paralyze prior to imaging [21–23].

The exact mechanism of action of 1-phenoxy-2-propanol (1P2P) remains unknown, but it has been shown to eliminate action potentials in neurons subsequently resulting in reduced muscle contraction [24]. Despite lack of a clear mechanism, 1P2P exposure induces ATP depletion in *C. elegans*, and both ATP depletion and collapse of mitochondrial membrane potential in Neuro2a cells [25]. Despite limited evidence, these data suggest mitochondria as a target for the mechanism of 1P2P. Mammalian toxicology evaluations find no risk of skin irritation and low levels of mucosal irritation suggesting low overt toxicity in mammalian models; however, mitochondrial endpoints were not evaluated [26].

Finally, cold shock is used in combination with physical impediments to movement such as polystyrene beads. Exposure to cold temperature slows metabolic processes in the worm, reducing movement. Cold shock at 2 C was only lethal after 12-hour exposures or longer, though cold shocked worms show decreased gonad size, loss of pigmentation, and increased vulval abnormalities after recovery [23]. Cold shock at 4 °C for 16 hours is capable of inducing blebbing of dopaminergic neurons, suggesting it may induce dopaminergic neurodegeneration [27]. As most paralytic uses of cold shock are far shorter in duration, it is unclear whether these effects occur in this timeframe.

Previous work has reviewed the time required to induce paralysis, recovery from paralysis, and induction of a number of stress-responsive genes via GFP reporter constructs after exposure to 1P2P, sodium azide, levamisole HCl, and cold shock [12]. However, the real-time impacts of these and additional paralytics on more dynamic reporters have not been assessed.

Here, we investigate how quickly the common anesthetics sodium azide, levamisole HCl, 1P2P, 2,3-butadione monoxime (2,3-BDM), and 4 °C cold shock induce paralysis, how quickly the worms can recover from exposure, and the effects of the treatments on two key biochemical parameters measured by fluorescent reporters: cellular redox tone, in this case assessed as the oxidation status of the glutathione pool of all cells (roGFP) [28] and relative ATP levels in muscle cells (QUEEN-2m) [8,29]. We report that all chemical paralytics, though not acute cold shock, induce changes to either redox tone or ATP levels at high doses typically used for effective paralysis and imaging, and highlight the need for tailoring anesthetic use to the experimental questions being used, or developing more viable non-chemical alternatives.

## Methods

### *C. elegans* strains and culturing

*C. elegans* were grown and maintained at 20 C on 10 cm K-agar plates seeded with OP50 *Escherichia coli* [30]. To synchronize worms' growth and development for experiments, adult worms were transferred to a fresh plate and allowed to lay eggs for two hours. After this time, all worms were rinsed from the plate until only the eggs remained. Eggs were allowed to mature for 72 hours before use in experiments (day 1 of adulthood). Strains used in this study were JV2

expressing a ribosomal promoter-driven cytosolic roGFP expressed in most or all cells; GA2001, expressing a myosin promoter-driven cytosolic Queen-2m expressed in muscle cells; and N2 Bristol (wild-type); all of which were obtained from the *C. elegans* Genome Center.

## Drugs and concentrations

All paralytics were prepared in K-medium [30] at the following concentrations: 10mM, 100mM, and 500mM sodium azide, 50 mM, 100mM, 200 mMand 300mM 2,3-butanedione monoxime, 0.5 mM, 1mM, 2 mM, and 3mM levamisole HCl, and 0.1%, 0.25%, 0.5% and 1.0% 1-phenoxy-2-propanol. The cold shock temperature was 4°C. Concentrations were selected based on commonly used doses in cited *C. elegans* literature. If only one dose is commonly used, such as 1% for 1P2P, a half dose was examined to provide a wider scope of evaluation for levels without off target effects.

## Assessment of paralysis

To assess paralysis, ten 20 μL drops of the given drug and concentration were placed in a small petri dish. One N2 worm was transferred into each drop by flame sterilized platinum pick. The time it was placed was noted, and the worm was checked every 60 seconds for the first five minutes, and then every five minutes until the worm was paralyzed, or until thirty minutes had passed. Paralysis was defined as no movement when the worm was touched with a pick. The time between when the worm was first placed in the drug droplet and when it reached paralysis was recorded. Time to 50% paralysis was determined by performing a Cumulative Gaussian non-linear fit in GraphPad Prism 9.3.0. The mean and standard deviation provided are best-fit values. The number of biological replicates for each treatment is as follows: Control = 6, 0.5 mM Levamisole = 3, 1 mM Levamisole = 6, 2 mM Levamisole = 3, 3 mM Levamisole = 6, 10 mM Sodium azide = 3, 100 mM Sodium Azide = 3, 500 mM sodium azide = 3, 0.1% 1P2P = 3, 0.25% 1P2P = 3, 0.5% 1P2P = 6, 1% 1P2P = 6. 50 mM 2,3-BDM = 3, 100mM 2,3-BDM = 6, 200 mM 2,3-BDM = 3, 300 mM 2,3 BDM = 6, 4°C cold shock = 3.

## Assessment of recovery from paralysis

To quantify recovery of the worms from paralysis by each agent, twenty worms were transferred into a 1.5 mL Eppendorf tube containing 400 μL of the given drug and concentration. After a 30-minute exposure, the drug was aspirated from the tube, and the worms were rinsed with K-medium three times. The worms were centrifuged for thirty seconds at 2200 rpm between each rinse to minimize loss of worms. In the event the worms stuck to the sides of the tube, 10 μL of 0.1% triton were added to the solution before rinsing.

After washing, exposed worms were placed onto a 10 cm K-agar plate seeded with OP50 *E. coli*. The worms were then observed every fifteen minutes for two hours. With each observation, the number of worms that moved when touched with a pick, or were moving on their own, was recorded. The number of biological replicates for each treatment is as follows: Control = 6, 0.5 mM Levamisole = 4, 1 mM Levamisole = 7, 2 mM Levamisole = 4, 3mM Levamisole = 7, 10 mM Sodium azide = 3, 100 mM Sodium Azide = 3, 500 mM sodium azide = 3, 0.1% 1P2P = 4, 0.25% 1P2P = 4, 0.5% 1P2P = 7, 1% 1P2P = 7, 50 mM 2,3-BDM = 4, 100mM 2,3-BDM = 7, 200 mM 2,3-BDM = 4, 300 mM 2,3 BDM = 7, 4°C cold shock = 3.

## Assessment of cytosolic redox tone and ATP levels

Synchronized adult JV2 or GA2001 worms were assessed using a FLUOstar Omega microplate reader (BMG Labtech) with each well containing 1000 worms suspended in 100 uL of K-

medium. Just before insertion into the plate reader, 100 μL of K-medium (control), 2X anesthetic, or $H_2O_2$ was added to the wells (final concentration of 1X). Fluorescence was recorded every 5 minutes for 35 minutes at 405 nm and 485 nm excitation wavelengths with a 520 nm emission wavelength.

To analyze the data, the average fluorescence of blank wells corresponding to each treatment was subtracted from the fluorescence of each corresponding well containing worms at each wavelength, which could then be converted to ratios of one excitation wavelength to another. This process was completed for each time point and each treatment group. For JV2 (roGFP) ratios are reported as oxidized: reduced, representing 485 nm excitation:405 nm excitation. For GA2001 (Queen-2m) ratios are reported as 405 nm:485 nm, representing ATP-bound:unbound protein. All experiments conducted with GA2001 worms (Queen-2m) were measured at ambient room temperature (25–26 C). Three biological replicates were assessed for redox tone for each treatment, with a final N for each groups as follows: Control = 7, 1 mM Levamisole = 13, 3mM Levamisole = 13, 10 mM Sodium azide = 13, 100 mM Sodium Azide = 13, 500 mM sodium azide = 12, 0.5% 1P2P = 9,1% 1P2P = 9, 100mM 2,3-BDM = 12, 300 mM 2,3 BDM = 12, 4˚C cold shock = 9, $H_2O_2$ = 14. Three biological replicates were assessed for relative ATP level for each treatment, with a final N for each groups as follows: Control = 8, 1 mM Levamisole = 16, 3mM Levamisole = 16, 10 mM Sodium azide = 12, 100 mM Sodium Azide = 8, 500 mM sodium azide = 10, 0.5% 1P2P = 12, 1% 1P2P = 13, 100mM 2,3-BDM = 12, 300 mM 2,3 BDM = 12, 4˚C cold shock = 9.

### Rotenone exposure

Synchronized adult JV2 or GA2001 worms were transferred to a 50 mL conical containing 1% DMSO (vehicle) or 20 uM Rotenone and OP50 *E. coli* in K+ medium and shaken on an orbital shaker for 1 hour. Worms were rinsed 3 times with K-medium, transferred to a 96 well plate, and assessed for cellular redox tone or relative ATP levels as described above. Three biological replicates were assessed for redox tone for each treatment, with n = 3 per replicate for a final n = 9 for each treatment group.

### Statistical analysis

Statistical analysis for all data was completed with GraphPad Prism 9.3.0. All data were assessed by a Shapiro-Wilks Normality test, followed by a 2-way Kruskal-Wallis as no data were normally distributed. Post-hoc tests consisted of Dunnett's multiple comparisons tests. P<0.05 was considered statistically significant, and all error bars reflected the standard error of the mean. Key statistical comparisons are described in the Results section, and full statistical analysis p-values are listed in S1–S4 Tables.

## Results

### Sodium azide rapid paralyzes while decreasing both ATP levels and redox tone

In this study we examined sodium azide at three concentrations: 10 mM, 100 mM, and 500 mM, and report that all sodium azide concentrations induce 50% population paralysis in less than a minute, and 100% by 30 minutes (Fig 1A). Though 10 mM and 100 mM exposures are 76.6% and 85.0% recoverable within 2 hours, this decreases dramatically to 1.8% at 500 mM (Fig 1A). All concentrations of sodium azide result in a rapid depletion of ATP in muscle cells that remains stable for at least 30 minutes (Fig 1B), consistent with the role of azide as an inhibitor of oxidative phosphorylation [9,31]. Similarly, all concentrations of sodium azide

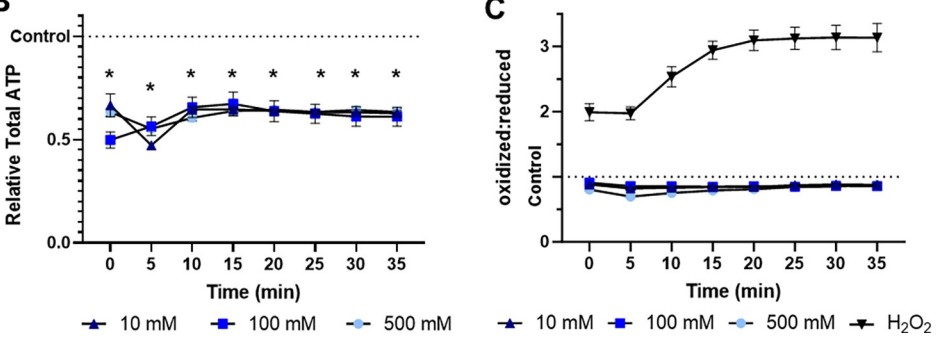

| Dose | Time to 50% Population Paralysis (min) | | Maximum Paralysis Achieved (% at 30 min) | | Percent Recovered within 2 hr (%) | |
|---|---|---|---|---|---|---|
| | Mean | Std Dev | Mean | Std Error | Mean | Std Error |
| 10 mM | 0.84 | 0.19 | 100.00 | 0.00 | 76.37 | 17.35 |
| 100 mM | 0.74 | 0.14 | 100.00 | 0.00 | 85.05 | 9.24 |
| 500 mM | 0.51 | 0.06 | 100.00 | 0.00 | 1.75 | 1.75 |

**Fig 1. Sodium azide induces rapid paralysis, ATP depletion, and briefly decreases glutathione pool oxidation. (A)** The time required for 10 mM, 100 mM, and 500 mM sodium azide to induce paralysis, the maximum percentage of worms paralyzed and the percentage that recover within 2 hours was determined.**(B)** Relative total muscle cell ATP levels were quantified for 35 minutes using the fluorescent reporter p*myo-3*::Queen-2m after exposure to 10 mM, 100 mM, and 500 mM sodium azide respectively. Statistical analysis was performed by two-way Kruskal-Wallis, with Dunnett's post-hoc. **(C)** Redox tone of the cytosolic glutathione pool was assessed through a reduction:oxidation sensitive GFP every 5 minutes for 35 minutes after exposure to 10 mM, 100 mM, and 500 mM sodium azide respectively. Statistical analysis was performed by two-way Kruskal-Wallis, with Dunnett's post-hoc.

cause a temporary decrease in cytosolic redox tone (i.e., more reduced state) of the glutathione pool (Fig 1C).

## Levamisole HCl induces paralysis and ATP depletion without alteration to cellular redox tone

Levamisole HCl is slightly slower to paralyze than sodium azide, requiring 3.04 and 1.92 minutes to achieve 50% population paralysis at 1 mM and 3 mM respectively (Fig 2A). It was also efficient as 88.77% and 93.33% of the population was paralyzed within 30 minutes. No individuals recovered within 2 hours, in agreement with previous reports stating that over 4 hours are required for recovery [12]. To ensure the lack of recovery did not represent lethality, recovery was assessed at 24 hours as well. At the highest dose recovery was 83.17%, with all other doses within a standard deviation of 100% (Fig 2A). Despite inducing ATP depletion, unlike azide, levamisole did not modify cytosolic redox tone (Fig 2B and 2C).

## 1P2P exposure causes high oxidative stress and ATP depletion

1P2P, at 0.5% and 1.0%, rapidly paralyzes worms in less than 7 and 4 minutes and achieves 98.49 and 100% paralysis respectively (Fig 3A). However, it induces ATP depletion that progressively increases over time, and high elevation of the cytosolic redox tone, indicating an increasingly oxidative cellular environment (Fig 3A and 3B). Notably, 73.47% and 45.00% of the nematodes tested recover within 2 hours at 0.5% and 1.0% doses; only 76.30 and 57.84% recovery at 24 hours suggesting that worms that do not recover within 2-hours are likely dead (Fig 3A).

| Dose | Time to 50% Population Paralysis (min) | | Maximum Paralysis Achieved (% at 30 min) | | Percent Recovered within 2 hr (%) | | Percent Recovered within 24 hr (%) | |
|---|---|---|---|---|---|---|---|---|
| | Mean | Std Dev | Mean | Std Error | Mean | Std Error | Mean | Std Error |
| 0.5 mM | N/A | N/A | 97.06 | 2.94 | 11.43 | 6.09 | 98.33 | 1.67 |
| 1 mM | 3.04 | 4.21 | 88.77 | 5.12 | 0.0 | 0.0 | 97.50 | 2.50 |
| 2 mM | 0.84 | 0.09 | 100.0 | 0.0 | 0.0 | 0.0 | 100.00 | 0.00 |
| 3 mM | 1.92 | 3.07 | 93.33 | 4.71 | 0.0 | 0.0 | 83.17 | 13.20 |

**Fig 2. Levamisole HCl induces paralysis and mild ATP depletion, without altering glutathione pool oxidation. (A)** The time required for 1 mM and 3 mM Levamisole HCl to induce paralysis, the maximum percentage of worms paralyzed and the percentage that recover within 2 hours was determined. **(B)** Relative total muscle cell ATP levels were quantified for 35 minutes using the fluorescent reporter p*myo-3*::Queen-2m after exposure to 1 mM and 3 mM Levamisole HCl respectively. Statistical analysis was performed by two-way Kruskal-Wallis, with Dunnett's post-hoc. **(C)** Redox tone of the cytosolic glutathione pool was assessed through a reduction:oxidation sensitive GFP every 5 minutes for 35 minutes after exposure to 1 mM and 3 mM Levamisole HCl respectively. Statistical analysis was performed by two-way Kruskal-Wallis, with Dunnett's post-hoc.

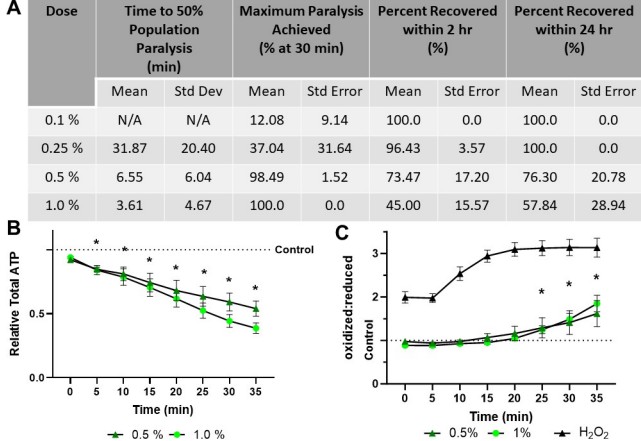

| Dose | Time to 50% Population Paralysis (min) | | Maximum Paralysis Achieved (% at 30 min) | | Percent Recovered within 2 hr (%) | | Percent Recovered within 24 hr (%) | |
|---|---|---|---|---|---|---|---|---|
| | Mean | Std Dev | Mean | Std Error | Mean | Std Error | Mean | Std Error |
| 0.1 % | N/A | N/A | 12.08 | 9.14 | 100.0 | 0.0 | 100.0 | 0.0 |
| 0.25 % | 31.87 | 20.40 | 37.04 | 31.64 | 96.43 | 3.57 | 100.0 | 0.0 |
| 0.5 % | 6.55 | 6.04 | 98.49 | 1.52 | 73.47 | 17.20 | 76.30 | 20.78 |
| 1.0 % | 3.61 | 4.67 | 100.0 | 0.0 | 45.00 | 15.57 | 57.84 | 28.94 |

**Fig 3. 1-Phenoxy-2-propanol exposure results in rapid paralysis, severe ATP depletion, and swift oxidation of the glutathione pool. (A)** The time required for 0.5% and 1.0% 1-phenoxy-2-propanol to induce paralysis, the maximum percentage of worms paralyzed and the percentage that recover within 2 hours was determined. **(B)** Relative total muscle cell ATP levels were quantified for 35 minutes using the fluorescent reporter p*myo-3*::Queen-2m after exposure to 0.5% and 1.0% 1-phenoxy-2-propanol respectively. Statistical analysis was performed by two-way Kruskal-Wallis, with Dunnett's post-hoc. **(C)** Redox tone of the cytosolic glutathione pool was assessed through a reduction:oxidation sensitive GFP every 5 minutes for 35 minutes after exposure to 0.5% and 1.0% 1-phenoxy-2-propanol respectively. Statistical analysis was performed by two-way Kruskal-Wallis, with Dunnett's post-hoc.

| Dose | Time to 50% Population Paralysis (min) | | Maximum Paralysis Achieved (% at 30 min) | | Percent Recovered within 2 hr (%) | | Percent Recovered within 24 hr (%) | |
|---|---|---|---|---|---|---|---|---|
| | Mean | Std Dev | Mean | Std Error | Mean | Std Error | Mean | Std Error |
| 50 mM | N/A | N/A | 5.56 | 2.94 | 97.50 | 2.50 | 100.0 | 0.0 |
| 100 mM | 18.90 | 11.73 | 75.91 | 8.93 | 78.06 | 13.50 | 75.00 | 25.00 |
| 200 mM | 8.74 | 3.23 | 100.0 | 0.0 | 43.18 | 25.55 | 78.13 | 17.95 |
| 300 mM | 3.79 | 2.54 | 100.0 | 0.0 | 0.00 | 0.00 | 27.50 | 24.28 |

**Fig 4. 2,3-Butadione monoxime treatment results in dose dependent paralysis, ATP depletion, and oxidation of the glutathione pool.** (**A**) The time required for 100 mM and 300 mM 2,3-butadione monoxime to induce paralysis, the maximum percentage of worms paralyzed and the percentage that recover within 2 hours was determined. (**B**) Relative total muscle cell ATP levels were quantified for 35 minutes using the fluorescent reporter p*myo-3*::Queen-2m after exposure to 100 mM and 300 mM 2,3-butadione monoxime respectively. Statistical analysis was performed by two-way Kruskal-Wallis, with Dunnett's post-hoc. (**C**) Redox tone of the cytosolic glutathione pool was assessed through a reduction:oxidation sensitive GFP every 5 minutes for 35 minutes after exposure to 100 mM and 300 mM 2,3-butadione monoxime respectively. Statistical analysis was performed by two-way Kruskal-Wallis, with Dunnett's post-hoc.

## 2,3-BDM is effective without clear toxicity at 100 mM, but lethal at 300 mM

2,3-BDM shows divergence between the examined concentrations. 100 mM results in slow paralysis of 75.91% of the population, with the majority recovering at 24 hours (75.00%, Fig 4A) and no significant effect on ATP levels or redox tone (Fig 4). Conversely, 300 mM 2,3-BDM rapidly depletes ATP levels and increases redox tone as it paralyzes 50% of the population in less than 4 minutes (Fig 4B and 4C). The increase in redox tone at 300 mM appears to be nearly as severe as treatment with 3% hydrogen peroxide, a dose lethal within 1 hour, although the rate of oxidation is slower. The 300 mM dose is also largely lethal, with only 27.50% of worms surviving 24-hours after a 1-hour exposure (Fig 4A).

## Acute cold shock does not paralyze, alter ATP levels, or change redox tone in worms

Finally, 30-minute 4°C cold shock alone is insufficient to paralyze more than 15% of the population (Fig 5A). This is anticipated as cold shock is typically utilized in combination with physical barriers or microfluidics. Importantly, cold shock did not detectably alter either muscular ATP levels or cytosolic glutathione pool redox tone, supporting its use over chemical paralytics from the perspective of disturbing these parameters (Fig 5A and 5B).

## Combined comparison of all methods

To facilitate direct comparison of all paralytic methods tested, we present graphs combining all replicates of all time-to-paralysis results (Fig 6A), time to recovery results (Fig 6B), effects on ATP (Fig 6C), and effects on redox state (Fig 6D).

## Levamisole and 1P2P induced paralysis modify detection of redox tone changes induced by rotenone exposure

To investigate how the use of the aforementioned paralytic agents could alter experimental outcomes, we examined the influence of paralytics on the outcome of a one-hour exposure to rotenone. One drug and dose each was selected as an agent that altered or did not alter redox tone in isolation. 3 mM levamisole HCl was selected for its lack of impact on redox tone and common use whereas 1% 1P2P was selected to determine if the rapid increase in oxidation would impact experimental conditions. One hour exposure to rotenone, in absence of a paralytic, results in a nearly 2-fold increase in the oxidized:reduced ratio of glutathione (Fig 7A–7C). When 3 mM levamisole is utilized as a paralytic, despite resulting in no difference on redox tone independently, it increases the magnitude of difference between vehicle and rotenone treated worms, largely driven by a more reduced redox tone in vehicle treated worms (Fig 7A, 7B and 7D). Similarly, and surprisingly, 1% 1P2P shows greater reduction in redox tone in both the vehicle and rotenone treated groups (Fig 7A, 7B and 7E). For both paralytics, rotenone exposure induces an increase in cellular redox tone with a relatively stable signal over the 30-minute measurement duration.

## 2,3-BDM and 1P2P alter the magnitude of change in relative ATP levels induced by rotenone exposure and produce an unstable signal over time

To investigate the impact of paralytics on experimental outcomes quantifying relative ATP levels, 100 mM 2,3-BDM was used for its lack of impact alone, whereas 1% 1P2P was selected for

**A**

| Dose | Time to 50% Population Paralysis (min) | | Maximum Paralysis Achieved (% at 30 min) | | Percent Recovered within 2 hr (%) | |
|------|------|------|------|------|------|------|
| | Mean | Std Dev | Mean | Std Error | Mean | Std Error |
| 4 C | 45.95 | 12.30 | 10.00 | 5.77 | 100.0 | 0.0 |

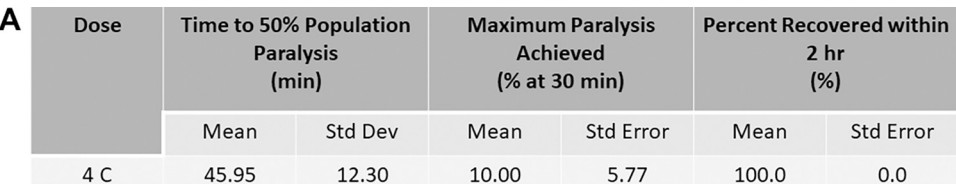

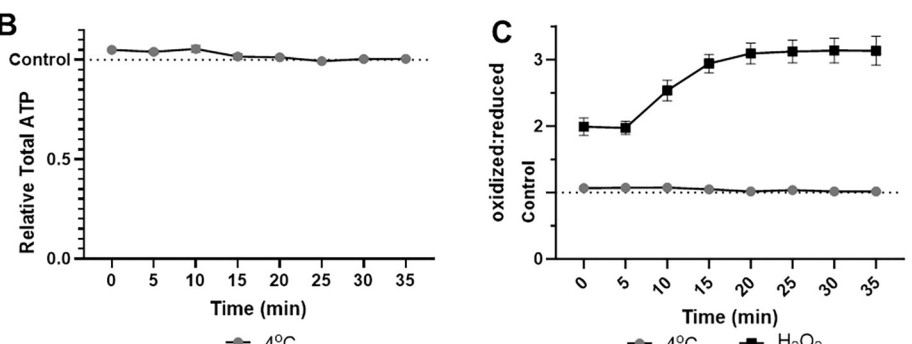

**Fig 5. 4°C Cold shock does not paralyze worms, alter muscular ATP level, or glutathione pool redox tone. (A)** The time required for 4°C cold shock to induce paralysis, the maximum percentage of worms paralyzed and the percentage that recover within 2 hours was determined. **(B)** Relative total muscle cell ATP levels were quantified for 35 minutes using the fluorescent reporter p*myo-3*::Queen-2m after exposure to 4°C cold shock. Statistical analysis was performed by two-way Kruskal-Wallis, with Dunnett's post-hoc. **(C)** Redox tone of the cytosolic glutathione pool was assessed through a reduction:oxidation sensitive GFP every 5 minutes for 35 minutes after exposure to 4°C cold shock. Statistical analysis was performed by two-way Kruskal-Wallis, with Dunnett's post-hoc.

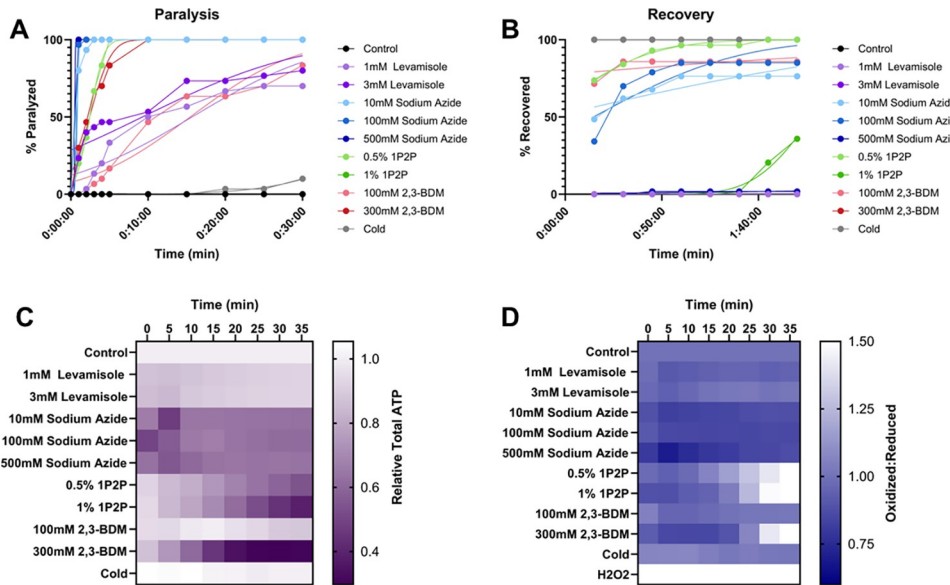

**Fig 6. No common chemical *C. elegans* anesthetic results in greater than 50% population paralysis without alteration to muscular ATP levels or cytosolic redox tone.** **(A)** The percentage of worms paralyzed over time after exposure to an anesthetic **(B)** The percentage of worms recovered from anesthetic exposure across a 2-hour monitoring window. **(C)** The evaluation of common anesthetics and doses to alter intra-muscular ATP levels was quantified every 5 minutes for 35 minutes with a p*myo-3*::Queen-2m fluorescent reporter. All anesthetics and doses were compared to non-treated controls at each time point through a 2-way Kruskal-Wallis with Dunnett's post hoc. **(D)** Redox tone of the cytosolic glutathione pool was assessed through a reduction:oxidation sensitive GFP every 5 minutes for 35 minutes after injection of common anesthetics. Statistical analysis was performed by two-way Kruskal-Wallis, with Dunnett's post-hoc. Each dose of each anesthetic was compared to controls at each timepoint.

its rapid depletion of ATP. In absence of a paralytic, rotenone exposure induced a stable, very small increase in relative ATP levels (Fig 8A–8C). Paralysis via 2,3-BDM resulted in a highly variable magnitude of change induced by rotenone over the 30-minute measurement period (Fig 8B), as well as a large decrease in both vehicle and rotenone treated groups over time (Fig 8C). Similarly, though to a lesser degree, 1P2P treatment resulted in a gradual decrease in the magnitude of rotenone's effect (Fig 8B) and relative ATP levels in both vehicle and treated groups (Fig 8D).

## Discussion

In this work, we report the impact of 4 chemical anesthetics and a 4°C cold shock on intramuscular cytosolic ATP levels and cytosolic redox tone of the glutathione pool (in all cells) using the dynamic *in vivo* reporters roGFP and QUEEN-2m in *C. elegans*. Assuming that equilibrium is rapidly reached between mitochondria and the cytosol for adenylate nucleotides and that glutathione oxidation in response to these anesthetics is similar in the cytosol and mitochondria, these cytosolic reporters allow insight into the impact of common anesthetics on mitochondrial function [32,33]. Investigating how long it took for worms to paralyze for each treatment revealed that all treatments studied, besides cold shock, lead to statistically significant paralysis in five minutes or less. 10 mM, 100 mM, and 500 mM sodium azide,1% 1-phenoxy-2-propanol, and 3 mM levamisole resulted in paralysis the fastest; however, 3 mM levamisole resulted in less ATP depletion and much less redox stress, making it the more appealing paralytic for rapid paralysis. Sodium azide still decreased redox tone and decreased ATP levels albeit less than 1P2P, though it remains an imperfect paralytic for mitochondrial or

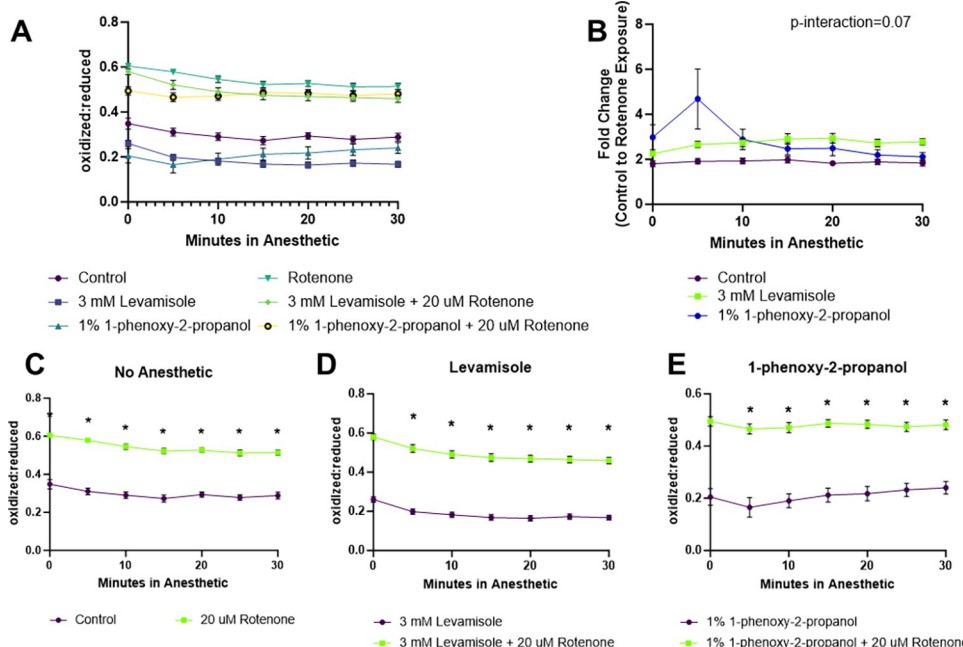

**Fig 7. Quantification of the impact of 3 mM levamisole and 1% 1P2P anesthetics on detection of rotenone induced alterations to cytosolic redox tone.** JV2 worms expressing cytosolic roGFP were exposed to 20 uM rotenone or 1% DMSO for 1-hour prior to assessment of redox tone. (A) The ratios of oxidized:reduced roGFP are reported with and without rotenone and anesthetic treatment. (B) To more clearly display how different anesthetics impact the detection of rotenone induced changes to cytosolic redox tone, the fold change between vehicle and rotenone treated worms is segregated by anesthetic. Fold change was quantified as the rotenone oxidized:reduced ratio divided by the average vehicle oxidized:reduced ratio at the same timepoint. The results of the raw ratios are also shown separately for (C) non-anesthetized worms, (D) 3 mM levamisole treated worms, and (E) 1% 1P2P treated worms.

redox related endpoints. The only chemical paralytic agent that, in isolation, does not decrease ATP or alter cytosolic redox tone is levels is 100 mM 2,3-BDM. However, it requires more than 15 minutes to achieve 50% population paralysis, decreasing its utility for measurements that need to be taken rapidly. Further, these results do not attest to precisely how an anesthetic may alter experimental outcomes.

The effects of many of these paralytics on ATP might have been anticipated based on their mechanisms of action, as described in the Introduction (although we again highlight that not all mechanisms of action are especially well understood). The effects on redox tone were not as predictable. Sodium azide may have resulted in a slightly more reduced redox state by hampering cellular utilization of reducing equivalents via oxidative phosphorylation, potentially resulting in a more reduced state of both the NAD+/NADH and NADP+/NADPH pools. Given minimal information regarding the mechanism of action of 1P2P, it is difficult to propose how this chemical might alter the cellular redox state, although we note that the result is consistent with previous reports of daf-16 activation [12] and loss of mitochondrial membrane potential [25] which could result from inhibition of oxidative phosphorylation and subsequent increased mitochondrial ROS production by mitochondrial electron leakage [33]. As suggested above, we hypothesize that the increased oxidation at 300 mM 2,3-BDM may reflect organismal death.

We cannot address the impact of every experimental design. However, to demonstrate the potential outcomes, we assessed how levamisole and 1P2P impacted detection of redox tone alterations induced by rotenone exposure. While an experiment performed with either paralytic would still demonstrate an increase in redox tone induced by rotenone, the magnitude or

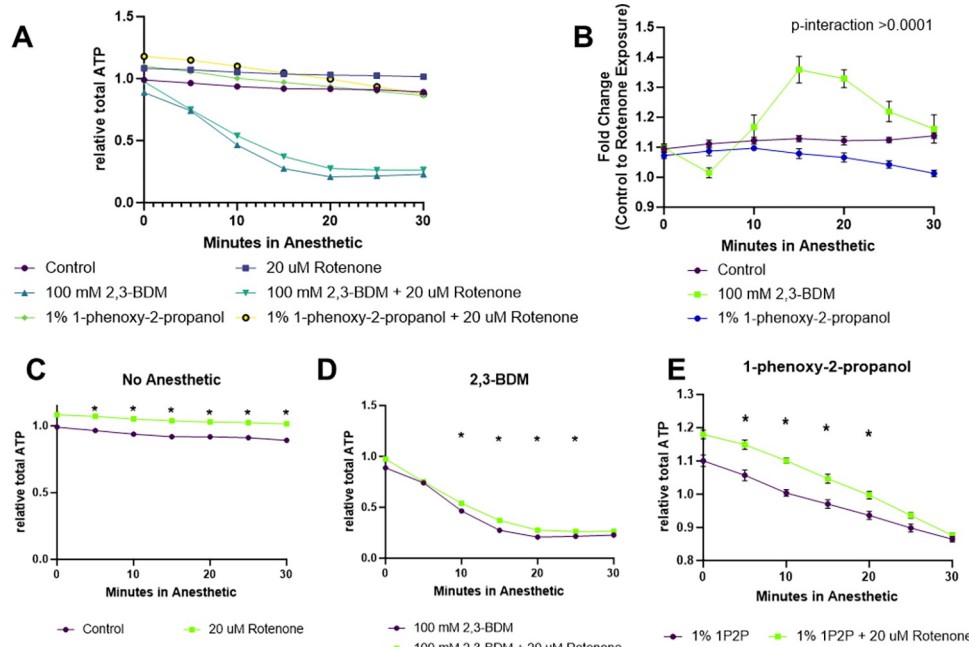

**Fig 8. Quantification of the impact of 100 mM 2,3-BDM and 1% 1P2P anesthetics on detection of rotenone induced alterations to relative ATP level.** GA2001 worms expressing QUEEN2-M in body wall muscle cells were exposed to 20 uM rotenone or 1% DMSO for 1-hour prior to assessment of relative ATP levels. (A) The raw ratios of ATP bound:unbound QUEEN2-M are reported with and without rotenone and anesthetic treatment. (B) To more clearly display how different anesthetics impact the detection of rotenone induced changes to relative ATP levels, the fold change between vehicle and rotenone treated worms is segregated by anesthetic. Fold change was quantified as the rotenone ATP bound:unbound ratio divided by the average vehicle ATP bound:unbound ratio at the same timepoint. The results of the raw ratios are also shown separately for (C) non-anesthetized worms, (D) 100 mM 2,3-BDM treated worms, and (E) 1% 1P2P treated worms.

difference and baseline measurements would be altered. Worse, when assessing 2,3-BDM and 1P2P impact on relative ATP levels in response to rotenone exposure, two major issues become apparent. First, neither paralytic results in a stable signal over time. Thus, in an experiment, if two treatment groups were paralyzed at the same time but imaged one after another, dramatically different results would be obtained solely as a result of the ATP depletion over time caused by 100 mM 2,3-BDM. As this change occurred within 30 minutes, this would also likely contribute to high individual variability within a treatment group. Second, the magnitude of difference between control and rotenone treated worms was inconsistent over time. This would result in decreased likelihood of detecting this small difference when the magnitude decreases or over-estimating it when the magnitude increases. The differences induced by 3 mM levamisole are small in magnitude, leading to the possibility that a lower dose may be safer. However, the lower dose of 1 mM levamisole induced a nearly identical decrease in muscle ATP levels within 5 minutes (Fig 2B) suggesting it would act in a similar manner to 3 mM. This, again, underscores how critical it is that individual researchers validate both their anesthetic of choice and dose within each unique experimental paradigm.

Notably, in the case of both ATP and redox tone, the paralytic agent selected for its lack of impact resulted in significant differences from un-paralyzed controls. Critically, the impact of a paralytic independently is not predictive of its impact on an experiment with a different experimental design. For example, it is surprising that 1P2P yielded a stable signal over time, however given the established impact of exercise on stress responses [34], it is likely the hour of swimming during rotenone exposure modified the response compared to worms that have

not exercised as in the experiments quantifying the impact of the paralytics alone. As every experiment will have different pre-treatments, genetic backgrounds, etc., it is impossible to predict exactly how these agents will interact in each scenario. Rather, we argue that it is critical for researchers to determine how a chosen paralytic method impacts their own scenario and account for this independently.

We also provide the first assessment of lethality induced by 1P2P, levamisole HCl, and 2,3-BDM. In the case of levamisole HCl, no paralyzed worms recover within 2 hours at doses higher than 0.5 mM. Conversely, 1P2P and 2,3-BDM were assessed for lethality to the combined high levels of ATP depletion and increased redox tone consistent with conditions observed during death. Notably, both 1P2P and 2,3-BDM induce high lethality at their highest doses. Considering that prior evaluations utilizing doses between 295–300 mM 2,3-BDM report required 30 minutes to a full hour before assessing their respective endpoint [21,22,35], it is possible the worms utilized in these assessments were dead or near death during assessment. Especially given the lack of mechanistic explanation for the lethality of 300 mM 2,3-BDM, we also assessed 3 lower doses, with lethality decreasing in a dose-responsive manner. Unfortunately, the dose at which no lethality was observed, 50 mM, is nearly useless as an anesthetic, paralyzing less than 6% of worms after 30 minutes. The lethality of an anesthetic that has been widely used highlights the need for pursuit of more ideal anesthetics or alternative methods.

The desire to pursue non-chemical anesthetics has led to a boom in the development of microfluidic devices, physical barriers such as polystyrene beads, and polymers that can be hardened after the addition of worms, such as BIO-133. As these methods rely on friction that the worms cannot overcome to move [36], loading the worms into narrow channels [37], or manipulation with acoustic waves [38,39], we predict that they are less likely to alter bioenergetics and redox tone. The ultraviolet (365 nm) light curable polymer BIO-133 [40,41] would be expected to induce DNA damage, though to our knowledge this has not been investigated. The use of methods that do not require hands-on dosing have also been employed in "lab-on-a chip" models to produce high-throughput readouts, pushing forward the ability to use *C. elegans* for chemical screening without anesthetic interference [42].

The lethality of 300 mM 2,3-BDM, as well as the impacts of other anesthetics on roGFP and Queen-2m readout, demonstrate the critical need to tailor anesthetic use to experimental methods and goals. Beyond on our evaluation, the impacts of the examined anesthetics may vary based on variation in exposure method, size of exposure groups (worms per amount of toxicant), and exposure conditions (liquid vs solid media), as shown in our evaluation of anesthetic impact on rotenone exposure read out in which the seemingly ideal 100 mM 2,3-BDM dramatically altered the outcome over time. Additionally, uptake of the drugs likely varies depending on life stage as the protective cuticle of the worm thickens with age, altering the amount of drug that can penetrate [43]. This was not an exhaustive study of all possible mitochondrial parameters, and the potential that the mitochondria are being impaired by the chemical in ways that the roGFP and Queen-2m assays are not sensitive to should be considered. However, our findings lay the groundwork for understanding which anesthetics are acceptable for bioenergetic and redox related endpoints and underscore the fact that anesthetics must be assessed within a given experimental design to ensure they will not influence the outcome.

## Supporting information

**S1 Table. Statistical analysis for time required to paralyze by each anesthetic P-values resulting from a two-way Kruskal Wallis test with Dunnet's post-hoc are reported for**

**percent paralysis when compared to the control over time.** Green shading indicates a statically significant p-value, while blue shading indicates a non-statistically significant p-value. (DOCX)

**S2 Table. Statistical analysis results for percent recovery from paralytic exposure.** P-values resulting from a two-way Kruskal Wallis test with Dunnet's post-hoc are reported for percent recovery when compared to the control over time. Green shading indicates a statically significant p-value, while blue shading indicates a non-statistically significant p-value. (DOCX)

**S3 Table. Statistical analysis of redox tone changes induced by each paralytic over time A two-way Kruskal-Wallis test performed provided p-values for oxidation state when compared to the control over time.** Green shading indicates a statically significant p-value, while blue shading indicates a non-statistically significant p-value. (DOCX)

**S4 Table. Statistical analysis of changes to relative muscle ATP levels over time A two-way Kruskal-Wallis test with Dunn's Post-hoc provided p-values for relative ATP levels when compared to the control over time.** Green shading indicates a statically significant p-value, while blue shading indicates a non-statistically significant p-value. (DOCX)

**S1 File.**
(CSV)

**S2 File.**
(CSV)

**S3 File.**
(CSV)

**S4 File.**
(CSV)

**S5 File.**
(CSV)

**S6 File.**
(CSV)

## Acknowledgments

We deeply appreciate the efforts of Sarah Seay for her maintenance of the laboratory supplies and facilities necessary to conduct the research presented here-in.

## Author Contributions

**Conceptualization:** Katherine S. Morton, Ashlyn K. Wahl, Joel N. Meyer.

**Data curation:** Ashlyn K. Wahl.

**Formal analysis:** Katherine S. Morton, Ashlyn K. Wahl.

**Funding acquisition:** Joel N. Meyer.

**Investigation:** Katherine S. Morton, Ashlyn K. Wahl.

**Methodology:** Katherine S. Morton, Ashlyn K. Wahl.

**Project administration:** Katherine S. Morton.

**Supervision:** Katherine S. Morton, Joel N. Meyer.

**Visualization:** Katherine S. Morton.

**Writing – original draft:** Katherine S. Morton, Ashlyn K. Wahl.

**Writing – review & editing:** Katherine S. Morton, Joel N. Meyer.

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
