## [Decision Letter · Decision Letter 0]

13 Nov 2023

PONE-D-23-30523The effect of common paralytic agents used for fluorescence imaging on redox tone and ATP levels in Caenorhabditis elegansPLOS ONE

Dear Dr. Morton,

Thank you for submitting your manuscript to PLOS ONE. After careful consideration, we feel that it has merit but does not fully meet PLOS ONE’s publication criteria as it currently stands. Therefore, we invite you to submit a revised version of the manuscript that addresses the points raised during the review process.

We look forward to receiving your revised manuscript.

Kind regards,

Agustín Guerrero-Hernandez

Academic Editor

PLOS ONE

“Some strains were provided by the Caenorhabditis Genetics Center, which is funded by NIH Office of Research Infrastructure Programs (P40 OD010440). This work was supported by National Institute of Health (P42ES010356, R01ES034270, 2R44OD024963, and T32ES021432).”

“This study was funded by the National Intitutes for Health (https://www.nih.gov/; JM-P42ES010356, R01ES034270,2R44OD024963, P40 OD010440, and T32ES021432). The funders had no role in study design, data collection and analysis, decision to publish, or preparation of the manuscript.”

Reviewers' comments:

Reviewer's Responses to Questions

**Comments to the Author**

1. Is the manuscript technically sound, and do the data support the conclusions?

Reviewer #1: Partly

Reviewer #2: Yes

2. Has the statistical analysis been performed appropriately and rigorously? 

Reviewer #1: Yes

Reviewer #2: Yes

3. Have the authors made all data underlying the findings in their manuscript fully available?

Reviewer #1: Yes

Reviewer #2: Yes

4. Is the manuscript presented in an intelligible fashion and written in standard English?

Reviewer #1: Yes

Reviewer #2: Yes

5. Review Comments to the Author

Reviewer #1: General comments

Caenorhabditis elegans is a well-established model for molecular biological and genetic experiments in vivo. The authors examined the effects of commonly used anesthetic methods on biochemical measurements such as cellular energetics and redox tone. Finally, the authors suggested that cold shock could be the best way when physical barriers were used in combination. Other anesthetic chemicals have merits and weak points regarding the lag time before the complete paralysis appears, the recovery time after the anesthesia, and energetics and redox in C. elegans. The findings are very informative for researchers who use worms as model animals. However, the authors provided only the nematodes' relative ATP levels and redox tone under the influence of each anesthetic method.

Usually, the experimental data are collected from the treated group and the control for the comparison, but both groups should be anesthetized similarly. For validation, this reviewer recommends the experiment where a representative chemical influencer to energetics or redox condition was administered to worms that received each one of ideal and inadequate anesthesia. Otherwise, the readers could think that significant differences should appear between the experimental and control groups, even under the worst anesthetic chemicals.

Specific comments

1. Lines 206-208. Please describe in detail. Fig 1C shows only one asterisk. Is this for the datum of 500 mM at 5 min or the data of total groups?

2. Line 243. “Fig 2A, 2B” should be corrected as “Fig 3B, 3C”, shouldn’t it?

3. Line 261. “Fig 4A” should be moved to just behind “(85.84%)”, shouldn’t it?

4. Line 263. “Fig 4B” should be moved as “Fig 4B, 4C” to just behind “3 minutes” on line 262, shouldn’t it?

5. Lines 262- 264. This reviewer cannot find the data to support this description in the manuscript.

6. Discussion. This reviewer is wondering if the authors briefly explain the pharmacological mechanisms of each anesthetic method from the viewpoints of total ATP, redox tone, and the paralytic effects.

Reviewer #2: Morton and colleagues test how commonly used paralytic agents, namely sodium azide, levamisole, 1-phenoxy-2-propanol, 2,3-BDM, and cold shock, affect time-to-paralysis, recovery, ATP levels, and redox tone in C. elegans. Importantly, the authors find that paralyzing treatments needed to achieve greater than 50% paralysis alter both redox tone and intramuscular cytosolic ATP levels. This work cautions experimenters to check the effects of paralyzing agents on final endpoints and optimize paralysis modes to avoid confounding results. Additionally, this work serves as a comparative resource for C. elegans biologists when choosing the appropriate paralytic agent and conditions for their specific question. While the data constitute a rather small minimal publishing unit, overall, the manuscript is scientifically valid and well-written. As PLOS ONE emphasizes scientific validity and rigor over perceived significance, I recommend that the manuscript be accepted with minor revisions.

Major Criticisms

• The paper could be improved through addition of positive controls for the ATP and redox experiments, especially done in the same plate as levamisole where only minor changes were seen. Other ATP reducers such as oligomycin or orthogonal methods such as starvation could be used as positive controls. The authors mention the use of H2O2 as a control for redox experiments, but those data are not plotted anywhere and are missing values in the Supporting Information roGFP.csv file.

• While we understand the authors do not intend to be exhaustive and picked the concentrations used from the literature, it might be relatively easy and informative to do paralysis and recovery curves for multiple doses of each chemical (especially in cases where only two concentrations were tested). It would be interesting to see, for a given exposure time, if there is a certain threshold concentration (no worms paralyzed vs. 100% worms paralyzed) or if greater concentrations lead to greater percentages of paralyzed worms. It would also boost the utility of this paper as a resource for the worm community, for example, scientists looking for a faster than 15-minute half-maximal paralysis rate achieved with 100 mM 2,3 BDM but a greater recovery rate than that achieved with 300 mM 2,3-BDM.

• Similar to the point above, data in Figure 5 could be elevated if different time exposures to cold shock or if some combinations of cold shock and chemical agent (as the authors point out is a commonly used approach) were tested.

• Overall, greater detail in either the Methods or the Figure Legends of the data being presented would be good, for example, number of total independent replicates done, if one or more replicates are represented in the heat maps, the temperatures for ATP experiments given the temperature-dependence of the QUEEN-2m sensor. Some replicate information is presented in the Supporting Information, but would be more accessible to the reader if also stated in the text.

6. PLOS authors have the option to publish the peer review history of their article (what does this mean?). If published, this will include your full peer review and any attached files.

Reviewer #1: No

Reviewer #2: No

---

## [Author Response · Author response to Decision Letter 0]

12 Feb 2024

Dear PLoS One:

We greatly appreciate the reviewers’ comments and have made every effort to respond thoroughly to each comment. Substantial additions to the manuscript are that we quantified how the redox tone and relative ATP level change as a result of rotenone exposure would be altered by anesthetics that do and do not have an impact independently. Both experiments significantly strengthen our conclusion, and we thank reviewers for improving this manuscript. Our point-by-point responses are in bold below, and we have uploaded both tracked-change and clean versions of the manuscript.

Thank you,

Katherine Morton, Ashlyn Wahl, and Joel Meyer

Point-by-point response:

“For validation, this reviewer recommends the experiment where a representative chemical influencer to energetics or redox condition was administered to worms that received each one of ideal and inadequate anesthesia. Otherwise, the readers could think that significant differences should appear between the experimental and control groups, even under the worst anesthetic chemicals.”

We have added two experiments to address this concern. As shown in Figures 7 and 8, we performed a rotenone exposure and examined how relative ATP and redox tone results compared when assessed with and without two anesthetics each. We demonstrate that even anesthetics not anticipated to have an impact would modulate the outcome of these experiments, with impacts ranging from alteration to the magnitude of difference between rotenone exposed and vehicle worms to unstable change over time in both treatment groups. While this does not comprehensively assess every anesthetic examined within the manuscript, we note that it is impossible to predict how each anesthetic would interact with all experimental designs and protocols. Rather, we aim to make the point that investigators must assess this outcome in their own experiments, and in the newly-added Figures 7 and 8 provide a proof of concept for what could happen. 

1. Lines 206-208. Please describe in detail. Fig 1C shows only one asterisk. Is this for the datum of 500 mM at 5 min or the data of total groups?

Thank you for pointing out the lack of clarity. This asterisk was removed as it was from an earlier version of the statistics that did not include the cold shock. In the updated statistics, the point in question is not statistically significant. Due to the close overlap in groups within the same graph, we opted to include the statistical comparisons for these data within the supplement, as referenced in line 190. 

2. Line 243. “Fig 2A, 2B” should be corrected as “Fig 3B, 3C”, shouldn’t it?

3. Line 261. “Fig 4A” should be moved to just behind “(85.84%)”, shouldn’t it?

4. Line 263. “Fig 4B” should be moved as “Fig 4B, 4C” to just behind “3 minutes” on line 262, shouldn’t it?

Thank you, these adjustments have been made as needed. 

5. Lines 262- 264. This reviewer cannot find the data to support this description in the manuscript.

Experimental data and explanation has been added to support this information in Figure 4.

6. Discussion. This reviewer is wondering if the authors briefly explain the pharmacological mechanisms of each anesthetic method from the viewpoints of total ATP, redox tone, and the paralytic effects.

We have described what is known about the mechanisms of action of each of these with regard to paralysis and possible bioenergetic alterations in the Introduction. We have added a small section to the Discussion regarding possible mechanisms of altered redox state.

The paper could be improved through addition of positive controls for the ATP and redox experiments, especially done in the same plate as levamisole where only minor changes were seen. Other ATP reducers such as oligomycin or orthogonal methods such as starvation could be used as positive controls. The authors mention the use of H2O2 as a control for redox experiments, but those data are not plotted anywhere and are missing values in the Supporting Information roGFP.csv file.

We apologize for the lack of clarity for the use of positive controls. For the redox data for each paralytic, the positive H2O2 control has been added to the graph. For the ATP data, we respectfully suggest that the results in Figure 1 for sodium azide provide a sufficient positive control.

While we understand the authors do not intend to be exhaustive and picked the concentrations used from the literature, it might be relatively easy and informative to do paralysis and recovery curves for multiple doses of each chemical (especially in cases where only two concentrations were tested). It would be interesting to see, for a given exposure time, if there is a certain threshold concentration (no worms paralyzed vs. 100% worms paralyzed) or if greater concentrations lead to greater percentages of paralyzed worms. It would also boost the utility of this paper as a resource for the worm community, for example, scientists looking for a faster than 15-minute half-maximal paralysis rate achieved with 100 mM 2,3 BDM but a greater recovery rate than that achieved with 300 mM 2,3-BDM.

We have added doses such that all drugs have at least 3 doses assessed for paralysis and recovery. ATP and redox measurements were not always tested in as many concentrations. In particular, if lower concentrations resulted in large departure from the controls, it seemed unnecessary to test the higher concentrations.

Similar to the point above, data in Figure 5 could be elevated if different time exposures to cold shock or if some combinations of cold shock and chemical agent (as the authors point out is a commonly used approach) were tested.

While we recognize the desire to examine a wider utility of cold shock, it is experimentally used with a wider variety in methods than we can reasonably examine. Rather, we include it here to demonstrate that independently it does not induce impacts, but this would need to be verified in other experimental designs. 

Overall, greater detail in either the Methods or the Figure Legends of the data being presented would be good, for example, number of total independent replicates done, if one or more replicates are represented in the heat maps, the temperatures for ATP experiments given the temperature-dependence of the QUEEN-2m sensor. Some replicate information is presented in the Supporting Information, but would be more accessible to the reader if also stated in the text.

We have added the number of biological replicates for each experiment and the temperature for Queen-2m measurements within the methods section.

---

## [Decision Letter · Decision Letter 1]

27 Feb 2024

PONE-D-23-30523R1The effect of common paralytic agents used for fluorescence imaging on redox tone and ATP levels in Caenorhabditis elegansPLOS ONE

Dear Dr. Morton,

Thank you for submitting your manuscript to PLOS ONE. After careful consideration, we feel that it has merit but does not fully meet PLOS ONE’s publication criteria as it currently stands. Therefore, we invite you to submit a revised version of the manuscript that addresses the points raised during the review process. There is still a minor clarification that need to be done in order to continue with the review process. Please, respond point by point or argue why does not apply in this particular case.

We look forward to receiving your revised manuscript.

Kind regards,

Agustín Guerrero-Hernandez

Academic Editor

PLOS ONE

Journal Requirements:

Reviewers' comments:

Reviewer's Responses to Questions

**Comments to the Author**

1. If the authors have adequately addressed your comments raised in a previous round of review and you feel that this manuscript is now acceptable for publication, you may indicate that here to bypass the “Comments to the Author” section, enter your conflict of interest statement in the “Confidential to Editor” section, and submit your "Accept" recommendation.

Reviewer #1: (No Response)

2. Is the manuscript technically sound, and do the data support the conclusions?

Reviewer #1: Yes

3. Has the statistical analysis been performed appropriately and rigorously? 

Reviewer #1: Yes

4. Have the authors made all data underlying the findings in their manuscript fully available?

Reviewer #1: Yes

5. Is the manuscript presented in an intelligible fashion and written in standard English?

Reviewer #1: Yes

6. Review Comments to the Author

Reviewer #1: General comments

The authors addressed well to the previous comments, adding new data and figures. However, this reviewer still has several queries below.

Specific comments

1. Line 210. What does this “d 100% by 30 minutesan” mean?

2. Line 213. The authors use “minutes” instead of “min” in other parts of this manuscript.

3. Lines 325-346. This reviewer cannot understand why the authors chose the 3 mM levamisole despite the fact that 2 mM of levamisole worked efficiently enough and showed fewer side effects in Figure 2.

4. Figures 7 and 8. The authors added these new informative figures, considering the previous comments. Unfortunately, this reviewer could find no description of how the authors calculated the fold changes of the Y-axis in panels B of both figures, reading the figure legend or “Materials and methods.”

5. Line 394. Please insert a space between “(12)” and “and.”

6. Lines 453-457. This reviewer admires the scientific manner of the authors and the modest description. However, this reviewer still needs to wonder why the authors hesitate to describe the conclusion. The data in Figures 2 and 6 likely indicate that 1- or 2-mM levamisole would be the first choice, subject to the authors providing the revised Figure 7, where 1- or 2-mM levamisole should be examined. Although 100 mM BDM looked like another suitable method in Figures 4 and 6, the new Figure 8 did not accept it. Cold shock looks safe and ideal but needs more time to get even the incomplete paralysis.

7. PLOS authors have the option to publish the peer review history of their article (what does this mean?). If published, this will include your full peer review and any attached files.

Reviewer #1: **Yes: **Yoshikazu Nishikawa

---

## [Author Response · Author response to Decision Letter 1]

12 Mar 2024

Dear PLoS One:

We greatly appreciate the reviewers’ comments and have made every effort to respond thoroughly to each comment. We have corrected previous grammatical and editing errors, and have added more clear explanations for our selection of doses in rotenone challenge experiments. Our point-by-point responses are in bold below, and we have uploaded both tracked-change and clean versions of the manuscript.

Thank you,

Katherine Morton, Ashlyn Wahl, and Joel Meyer

1. Line 210. What does this “d 100% by 30 minutesan” mean?

This was an editing error and has been corrected. 

2. Line 213. The authors use “minutes” instead of “min” in other parts of this manuscript.

We have corrected all use of “min” to “minutes” throughout the manuscript. 

3. Lines 325-346. This reviewer cannot understand why the authors chose the 3 mM levamisole despite the fact that 2 mM of levamisole worked efficiently enough and showed fewer side effects in Figure 2.

Thank you for pointing out the lack of clarity for the choice of the 3 mM dose. A range of doses for levamisole are explored in the literature ranging up to 5 mM depending on the circumstances of the investigation. We selected 3 mM as a dose closer to those commonly used in experiments, and because the impact on relative muscle ATP levels is nearly identical to the 1 mM dose, demonstrating that 1 mM levamisole would likely still result in impacts. We note that while it may appear at first glance that 2 mM is a “Goldilocks” concentration given fast, 100% paralysis, we suspect it is not actually as perfect as it might appear. It would be very surprising if it were to work so significantly better than both 1 and 3 mM in terms of paralysis; we think a more conservative and likely true interpretation of the full dose-response data set is that all three concentrations behave similarly, and the apparently superior paralytic effect at 2 mM is a random result.

We emphasize that our goal is not to supply the perfect dose and anesthetic. Rather, we aim to demonstrate the potential impacts of a commonly used anesthetic and dose to underscore the requirement for researchers to validate their use of any chemical anesthetic in the future. 

We have added sentences (lines 419-424) to clarify these points. 

Thank you for pointing out the lack of clarity for dose selection. 

4. Figures 7 and 8. The authors added these new informative figures, considering the previous comments. Unfortunately, this reviewer could find no description of how the authors calculated the fold changes of the Y-axis in panels B of both figures, reading the figure legend or “Materials and methods.”

Thank you for pointing out this lack of description. We have added the method used to calculate fold change into the figure description of both Figures 7 and 8. 

5. Line 394. Please insert a space between “(12)” and “and.”

Thank you, we have corrected this. 

6. Lines 453-457. This reviewer admires the scientific manner of the authors and the modest description. However, this reviewer still needs to wonder why the authors hesitate to describe the conclusion. The data in Figures 2 and 6 likely indicate that 1- or 2-mM levamisole would be the first choice, subject to the authors providing the revised Figure 7, where 1- or 2-mM levamisole should be examined. Although 100 mM BDM looked like another suitable method in Figures 4 and 6, the new Figure 8 did not accept it. Cold shock looks safe and ideal but needs more time to get even the incomplete paralysis.

Thank you, we deeply appreciate your compliment. We believe we have addressed the concerns regarding levamisole above. We have also added two sentences (lines 464-466 and 472-474) to more clearly describe our conclusions.

---

## [Decision Letter · Decision Letter 2]

1 Apr 2024

The effect of common paralytic agents used for fluorescence imaging on redox tone and ATP levels in Caenorhabditis elegans

PONE-D-23-30523R2

Dear Dr. Morton,

We’re pleased to inform you that your manuscript has been judged scientifically suitable for publication and will be formally accepted for publication once it meets all outstanding technical requirements.

Kind regards,

Agustín Guerrero-Hernandez

Academic Editor

PLOS ONE

Additional Editor Comments (optional):

Reviewers' comments:

Reviewer's Responses to Questions

**Comments to the Author**

1. If the authors have adequately addressed your comments raised in a previous round of review and you feel that this manuscript is now acceptable for publication, you may indicate that here to bypass the “Comments to the Author” section, enter your conflict of interest statement in the “Confidential to Editor” section, and submit your "Accept" recommendation.

Reviewer #1: All comments have been addressed

2. Is the manuscript technically sound, and do the data support the conclusions?

Reviewer #1: Yes

3. Has the statistical analysis been performed appropriately and rigorously? 

Reviewer #1: Yes

4. Have the authors made all data underlying the findings in their manuscript fully available?

Reviewer #1: Yes

5. Is the manuscript presented in an intelligible fashion and written in standard English?

Reviewer #1: Yes

6. Review Comments to the Author

Reviewer #1: (No Response)

7. PLOS authors have the option to publish the peer review history of their article (what does this mean?). If published, this will include your full peer review and any attached files.

Reviewer #1: **Yes: **Yoshikazu Nishikawa
